# MULTI-FREQUENCY FUSION FOR ROBUST VIDEO FACE FORGERY DETECTION

## ABSTRACT

Current face video forgery detectors use wide or dual-stream backbones. We show that a single, lightweight fusion of two handcrafted cues can achieve higher accuracy with a much smaller model. Based on the Xception baseline model (21.9 million parameters), we build two detectors: LFWS, which adds a 1x1 convolution to combine a low-frequency Wavelet-Denoised Feature (WDF) with the phase-only Spatial-Phase Shallow Learning (SPSL) map, and LFWL, which merges WDF with Local Binary Patterns (LBP) in the same way. This extra module adds only 292 parameters, keeping the total at 21.9 million—smaller than F3Net (22.5 million) and less than half the size of SRM (55.3 million). Even with this minimal overhead, the fused models increase the average area under the curve (AUC) from 74.8% to 78.6% on FaceForensics++ and from 70.5% to 74.9% on DFDC-Preview, gains of 3.8% and 4.4% over the Xception baseline. They also consistently outperform F3Net, SRM, and SPSL in eight public benchmarks, without extra data or test-time augmentation. These results show that carefully paired, handcrafted features, combined through the lightweight fusion block, can provide state-of-the-art robustness at a significantly lower cost. Our findings suggest a need to reevaluate scale-driven design choices in face video forgery detection.

## 1 INTRODUCTION

Powerful generative models, such as GANs and diffusion models, have made it easy to create convincing deepfakes; however, distinguishing them from real images remains a significant challenge (Schwarz et al., 2021; Dzanic et al., 2020). Many recent detectors look for *high-frequency* inconsistencies. Local Binary Patterns (LBP) find microtexture anomalies (Ojala et al., 2002), Fourier-based methods reveal spectral artifacts (Dzanic et al., 2020), and Spatial-Phase Shallow Learning (SPSL) isolates phase distortions (Liu et al., 2021). Wavelet-based techniques, in contrast, show that *low-frequency* structural cues are also useful (Wolter et al., 2021). However, most existing methods focus on just one frequency band or require complex, full-scale decompositions, which makes them less practical for real-time use and limits their ability to generalize.

To solve this problem, we introduce a multi-frequency fusion framework. Our method extracts a compact Wavelet Denoised Feature (WDF) channel to keep low-frequency structure, then combines it with targeted high-frequency information from Local Binary Patterns (LBP) or Spatial-Phase Shallow Learning (SPSL). We propose two lightweight fusion variants:

- **LFWL**: *L*ightweight *F*usion of *W*DF with *L*BP

- **LFWS**: *L*ightweight *F*usion of *W*DF with *S*PSL

The fusion blocks integrate low- and high-frequency feature maps into a single additional channel. This design enables any backbone architecture to process a four-channel input with minimal computational overhead.

We show consistent improvements over single-frequency baselines and earlier fusion methods on eight public benchmarks, including FaceForensics++, Celeb-DF v1/v2, and DFDC. Our approach achieves state-of-the-art cross-domain robustness with fewer parameters.

## 2 RELATED WORK

### 2.1 CNN-BASED DETECTORS

Early deepfake detectors standard Convolutional Neural Networks, such as MesoNet (Afchar et al., 2018), ResNet (He et al., 2016), EfficientNet-B4 (Tan & Le, 2019), and Xception (Chollet, 2017), which were trained on face-swap datasets. While these models performed well on their original data, they often struggled to adapt to new manipulation styles and datasets.

### 2.2 FREQUENCY-AWARE APPROACHES

Recent studies have moved away from using only raw RGB pixel values and now use specially designed frequency-based features to detect deepfakes. F3Net fuses multiscale Fourier bands to expose synthesis errors (Qian et al., 2020); SRM adopts steganalysis filters to highlight residual noise (Luo et al., 2021); SPSL keeps only phase information to reveal boundary distortions (Liu et al., 2021). All three are benchmarked in DeepfakeBench (Yan et al., 2023) for reproducibility.

### 2.3 BENCHMARKS

To investigate how training data diversity affects cross-domain robustness, we employed two experimental protocols using the same neural network architecture.

- **FaceForensics++** (Rössler et al., 2019): A widely adopted benchmark with four classic manipulation styles.
- **DFDC Preview (DFDCP)** (Dolhansky et al., 2019): A 5 k video subset selected for its rich demographic and scene variation, testing the model's ability to generalize beyond familiar manipulations.

We did not utilize the full Deepfake Detection Challenge (DFDC) dataset (Dolhansky et al., 2020) for training, which comprises over 100,000 videos, as it requires excessive computing power for repeated experiments.

### 2.4 GAP AND MOTIVATION

Frequency-based detectors are effective at identifying high-frequency artifacts but often overlook important low-frequency structural details. To address this, we combine a single Wavelet-Denoised low-frequency map (WDF) (Mallat, 1989) with a high-frequency map, either Local Binary Patterns (LBP) (Ojala et al., 2002) or SPSL (Liu et al., 2021), using a 1×1 convolution. We then add this fused channel to the RGB input, keeping the original neural network architecture unchanged. Across all tests, our compact multi-frequency fusion consistently improves accuracy, efficiency, and cross-dataset generalization over single-frequency or naïve-concatenation baselines.

## 3 PROPOSED METHOD

### 3.1 OVERVIEW OF THE ARCHITECTURE

Our proposed framework is built upon the Xception backbone to incorporate multiple handcrafted feature streams. As shown in Figure 1, the model takes as input:

- **RGB image**: normalized to the range $[-1, 1]$.
- **Handcrafted features**: extracted via various image-processing techniques, each also normalized to $[-1, 1]$.

We investigate two main strategies for combining these features with the RGB channels:

1. **Method 1 (Concatenation):** Direct concatenation of the handcrafted feature maps and RGB image, fed into the Xception backbone.

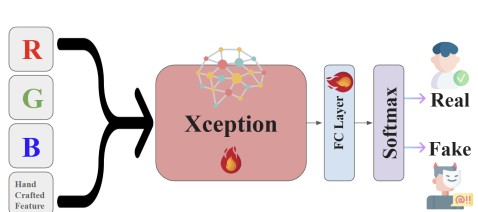

**Method 1: Concatenation Approach**

One channel handcrafted feature (e.g.,SPSL, LBP, WDF) is directly concatenated with the RGB image. The combined 4-channel input is passed to the backbone.

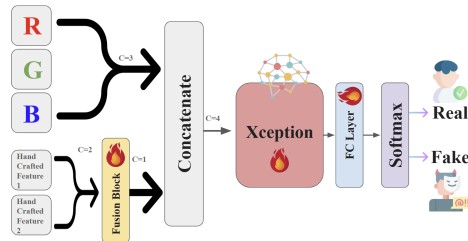

**Method 2: Learnable Fusion Block**

A lightweight, trainable module that fuses multiple handcrafted feature streams (e.g., SPSL, WDF, LBP) into a single-channel representation, which is then concatenated with the RGB image.

Figure 1: Comparison of the two feature integration strategies. Method 1 naively concatenates a handcrafted feature map with RGB inputs, while Method 2 fuses selected feature maps into a compact representation, improving both accuracy and efficiency.

2. **Method 2 (Learnable Fusion Block):** A lightweight, trainable fusion block that fuses two feature streams into a single representation map before concatenation with the RGB image.

All experiments are implemented on top of the DeepfakeBench framework (Yan et al., 2023), which we extend to support both Method 1 (handcrafted features: LBP, WDF) and Method 2 (learnable fusion features: LFWS, LFWL). Our implementation enables consistent training and evaluation of both baseline and proposed models across a unified interface.

After training, the fusion block can be frozen, allowing the fused features alone to be transferred to alternative backbones (e.g., MesoNet, EfficientNet), thereby enabling flexibility and scalability.

### 3.2 Handcrafted Feature maps and Normalization

We append three artefact-oriented channels to the RGB tensor:

- **Wavelet-Denoised Feature (WDF).** The grayscale frame is decomposed by a 3-level 2-D `db1` wavelet; all detail coefficients are zeroed and the low-frequency approximation is inverse–transformed, yielding a single coarse map that is re-scaled to $[-1, 1]$.
- **LBP.** We compute *uniform* Local Binary Patterns (LBP) with radius 1 and $P=8$ neighbors using `local_binary_pattern` from SKIMAGE, resulting in codes ranging from $0$ to $P+1 = 9$. To ensure stable, zero-centered inputs for CNN training, we linearly normalize the LBP codes to the range $[-1, 1]$. The normalization is defined as

$$x_{\text{norm}} = \frac{x}{P+2} \times 2 - 1, \tag{1}$$

  where the denominator $P+2$ slightly compresses the upper bound, helping center the distribution closer to zero and improving training stability.
- **SPSL.** Following Liu et al. (2021), we take the grayscale image, apply a 2-D FFT, and keep only the phase spectrum ($\angle X$), set the magnitude to 1, and perform an inverse FFT. The resulting real part is a *phase-only* image already bounded in $[-1, 1]$.

Consequently, the WDF, LBP, and RGB channels are *explicitly* normalised to $[-1, 1]$; the phase channel already lies in that range, so the tensors$(R, G, B, \text{WDF}, \text{LBP}, \text{SPSL})$ feed the models with balanced magnitudes.

### 3.3 Lightweight Fusion Block (Method 2)

Figure 2 illustrates the proposed module, which compresses two handcrafted maps into a single learned channel and concatenates it with RGB to form a 4-channel input for Xception.

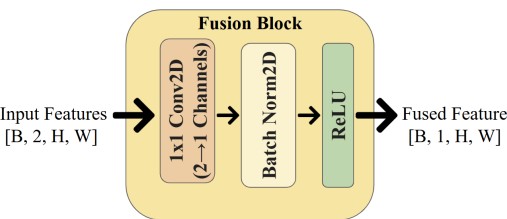

Figure 2: Architecture of the lightweight trainable fusion block. Handcrafted features (e.g., SPSL and WDF) are concatenated along the channel dimension and passed through a $1 \times 1$ convolution, followed by batch normalization and ReLU. The output is a single fused feature map, which is then concatenated with the RGB image.

**Design.** A $1 \times 1$ convolution mixes either WDF+SPSL (**LFWS**) or WDF+LBP (**LFWL**), followed by Batch Normalization and ReLU. This lightweight block introduces only two additional parameters per spatial location, stabilizes gradients, and couples low- and high-frequency cues.

**Deployment.** After training, mixer weights and BN statistics are frozen, so the block can be reused with other backbones at negligible cost.

**Advantages.** Reducing channel with Method 2 makes computation more efficient and helps the network learn a better representation instead of just combining features. Adding 292 extra weights, which is very small compared to the 21.9 million parameters in Xception (less than 0.0014 percent), leads to consistent accuracy improvements. This suggests that learnable pairwise fusion can detect important manipulation artifacts and reduce redundancy, which aligns with previous findings on channel efficiency. (Hu et al., 2018; Liang et al., 2020).

### 3.4 IMPLEMENTATION DETAILS

Below is a summary of the main training configuration parameters.

- **Pretrained Backbone:** Xception initialized with publicly available PyTorch weights (`xception-b5690688.pth`, as used in DeepfakeBench).
- **Data Augmentation:** Each transformation was applied independently with a probability of 0.5. These included horizontal flipping, random rotation (from -10° to +10°), Gaussian blur (kernel size 3 to 7), brightness and contrast adjustments (±10 %), and JPEG compression (quality 40 to 100).
- **Normalization:** All channels (RGB and handcrafted feature maps) scaled to $[-1, 1]$ using $\mu = 0.5$ and $\sigma = 0.5$.
- **Optimizer:** We use the Adam optimizer with a learning rate of $2 \times 10^{-4}$, momentum parameters $\beta_1 = 0.9$ and $\beta_2 = 0.999$, an epsilon of $1 \times 10^{-8}$, AMSGrad set to `false`, and a weight decay of $5 \times 10^{-4}$.
- **Training Schedule:** We trained the model for 10 epochs using a batch size of 32, saving checkpoints after each epoch.
- **Loss Function:** We use binary cross-entropy loss for all models except SRM, where Additive Margin Softmax (AM-Softmax) is applied to align with the original implementation.
- **Hardware and Software:** Training was performed on GPU-shared nodes (2× V100-32 GB, 120 GB RAM) using Python 3.8.19, PyTorch 1.12.0+cu113, torchvision 0.13.0+cu113, CUDA 11.3, and cuDNN 8.3.2 with acceleration enabled.
- **Manual Seed:** All experiments were conducted with the random seed fixed to 1024.

## 4 EXPERIMENTS AND RESULTS

This section presents the experimental setup, datasets, and evaluation protocols, followed by qualitative and quantitative analyses of the proposed method. Unless specified otherwise, all implementation

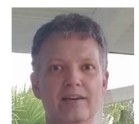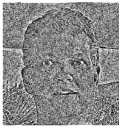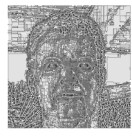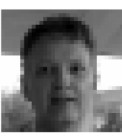

RAW       SPSL       LBP       WDF

Figure 3: Visual examples of handcrafted features extracted from a single image. From left to right: original RGB, Spatial-Phase Shallow Learning (SPSL), Local Binary Patterns (LBP), and Wavelet-Denoised Features (WDF). Each feature highlights different manipulation artifacts across spatial and frequency domains.

details, including optimizer, data augmentation, and normalization, are consistent with those described in Section 3.

### 4.1 DATASETS & SETUP

**Training.** Two identical networks are trained: one on FaceForensics++ (c23) (Rössler et al., 2019), The medium-compression version commonly used in benchmarks that includes four manipulation styles (DF, F2F, FS, and NT), and one on the DFDC Preview (DFDCP) (Dolhansky et al., 2019), which provides a broader range of demographics.

**Testing.** Both models are evaluated on eight public benchmarks: FaceForensics++ (c23) (Rössler et al., 2019), FaceShifter (Li et al., 2020a), Google DFD (DeepfakeDetection) (Google AI Blog, 2019), Celeb-DF v1 and v2 (Li et al., 2020b), DFDC (Dolhansky et al., 2020), DFDCP (Dolhansky et al., 2019), and UADFV (Li et al., 2018). The average area under the curve (AUC) is reported across all benchmarks, without separating within-domain and cross-domain results. This unified metric incorporates both straightforward and complex cases, accounts for uncertainty in data provenance, and facilitates fair comparison by avoiding double-counting of datasets.

All datasets are used with their official splits, ensuring a strict separation between training and testing; for instance, FF++ and DFDCP in the evaluation are drawn exclusively from their designated test sets. Each video is sampled into 32 frames, and train/test splits are made at the video level to prevent leakage, with models trained and evaluated on the extracted frames.

### 4.2 BASELINE ANALYSIS: FEATURE-ONLY INPUTS

We evaluate the discriminative power of each handcrafted feature by feeding individual features—Wavelet Denoised Features (WDF), Spatial-Phase Shallow Learning (SPSL), and Local Binary Patterns (LBP)—into an Xception backbone without concatenating raw RGB images.

- **WDF:** AUC = 0.9519
- **SPSL:** AUC = 0.9477
- **LBP:** AUC = 0.9388

WDF achieves the highest AUC, showing that resampling and compression artifacts provide informative signals in low-frequency regions. SPSL and LBP also yield strong performance, supporting the assertion that frequency- and phase-based features encode relevant cues for forgery detection. These findings provide a rationale for fusing complementary representations, such as WDF with SPSL or LBP, to capture a broader spectrum of artifacts. These baseline models were trained and evaluated on the FaceForensics++ dataset without data augmentation.

### 4.3 FUSION APPROACHES AND FINAL DETECTION PERFORMANCE

There are two fusion strategies, as detailed in Section 3 and illustrated in Figure 1. Method 1 utilizes direct concatenation, while Method 2 employs a lightweight fusion block. For Method 2, two fused models are evaluated: LFWS, which combines WDF and SPSL, and LFWL, which combines WDF and LBP.

Table 1: **Forgery detection AUC trained on FF++ (c23).** We highlight top 3 average AUC values in **blue bold**. The last row reports the relative improvement (%) in average AUC compared to the baseline Xception model, demonstrating the benefit of integrating handcrafted and fused features for cross-domain robustness.

| Dataset | Baseline CNNs | | | | Freq-based | | | Method 1 | | Method 2: Fusion | |
|---|---|---|---|---|---|---|---|---|---|---|---|
| | Xcept | Meso4 | RNet34 | EffB4 | f3net | SRM | SPSL | LBP | WDF | LFWS | LFWL |
| FaceForensics++ | 0.9805 | 0.7135 | 0.9665 | 0.9765 | 0.9791 | 0.9770 | 0.9789 | 0.9662 | 0.9778 | 0.9761 | 0.9741 |
| FaceShifter | 0.6248 | 0.6275 | 0.5506 | 0.6062 | 0.6062 | 0.5663 | 0.6629 | 0.6434 | 0.6581 | 0.6163 | 0.6150 |
| DFD | 0.8042 | 0.5524 | 0.7691 | 0.8535 | 0.8281 | 0.8111 | 0.8141 | 0.7983 | 0.8088 | 0.8241 | 0.8146 |
| Celeb-DF-v1 | 0.6238 | 0.6022 | 0.6607 | 0.7363 | 0.7052 | 0.7466 | 0.6734 | 0.7106 | 0.6860 | 0.7875 | 0.7188 |
| Celeb-DF-v2 | 0.6875 | 0.6080 | 0.6712 | 0.6684 | 0.7083 | 0.7345 | 0.6999 | 0.7216 | 0.7108 | 0.7548 | 0.7324 |
| DFDCP | 0.6764 | 0.5734 | 0.6355 | 0.6737 | 0.7259 | 0.7019 | 0.7249 | 0.7149 | 0.7244 | 0.6991 | 0.7345 |
| DFDC | 0.6862 | 0.5720 | 0.6932 | 0.6747 | 0.6938 | 0.6838 | 0.7024 | 0.6940 | 0.6838 | 0.7172 | 0.7072 |
| UADFV | 0.8966 | 0.8714 | 0.9099 | 0.9072 | 0.9043 | 0.9163 | 0.8805 | 0.8975 | 0.9167 | 0.9110 | 0.9325 |
| Avg AUC | 0.7475 | 0.6401 | 0.7321 | 0.7621 | 0.7689 | 0.7672 | 0.7671 | 0.7683 | 0.7708 | 0.7858 | 0.7786 |
| Improvement (%) | – | -10.74 | -1.54 | +1.46 | +2.14 | +1.97 | +1.96 | +2.08 | +2.33 | +3.83 | +3.11 |

The baseline models consist of standard convolutional neural networks (CNNs), including Xception, Meso-4 (MesoNet), ResNet-34, and EfficientNet-B4, as well as frequency-based models such as F3Net, SRM, and SPSL. Although SPSL utilizes feature concatenation as in Method 1, it is categorized with frequency-based models to maintain consistency with the original implementation and ensure fair comparison.

All handcrafted feature maps are normalized or rescaled to the range [-1,1] prior to fusion or concatenation. This preprocessing step ensures balanced training across feature representations.

### 4.3.1 TRAINED ON FACEFORENSICS++ (C23)

**Overall Results.** Table 1 reports AUC scores across eight benchmarks for models trained solely on FaceForensics++ (c23). The results include baseline CNNs, handcrafted feature-based variants, frequency-based models (F3Net, SRM, SPSL), and our proposed approaches (Method 1: concatenation; Method 2: lightweight fusion).

**Key Observations (FF++ Training).**

- **Fusion methods generalize best.** Among all models, Method 2 fusions demonstrate the strongest generalization. LFWS (fusing WDF + SPSL) achieves the highest average AUC of **0.7858**, outperforming every baseline and frequency-based method. LFWL (WDF + LBP) also ranks second with **0.7786**, followed closely by WDF (**0.7708**). These results show the complementary nature of handcrafted features when fused with a learnable module.

- **Notable improvements on challenging datasets.** On cross-domain benchmarks, including Celeb-DF-v1 and Celeb-DF-v2, LFWS achieves AUC scores of 0.7875 and 0.7548, respectively. These values are significantly higher than those of all baseline models, including SRM and F3Net. The results indicate that the fused representations provide increased robustness to dataset shifts and previously unseen manipulations.

- **Method 2 outperforms Method 1.** Direct concatenation in Method 1 yields only modest improvements over the baselines and does not achieve the performance of Method 2. This performance gap demonstrates the benefit of a lightweight, trainable fusion block that compresses and non-linearly refines feature interactions, rather than simply stacking them.

- **Baseline CNNs and frequency-based methods plateau.** While SRM (0.7672) and F3Net (0.7689) perform competitively, they do not surpass the handcrafted + fusion combinations. Among CNN baselines, EfficientNet-B4 (0.7621) is the best, though it still falls short of the fusion-based approaches.

### 4.3.2 TRAINED ON DFDCP

**Overall Results.** Table 2 lists AUC scores for models trained on DFDCP (same number of videos as FF++ but broader real-world variability) and evaluated on eight benchmarks.

**Analysis (DFDCP Training).**

Table 2: **Forgery detection AUC trained on DFDCP.** We highlight the top 3 average AUC values in **blue bold**. The last row reports the relative improvement (%) in average AUC compared to the baseline Xception model, demonstrating the benefit of integrating handcrafted and fused features for cross-domain robustness.

| Dataset | Baseline CNNs | | | | Freq-based | | | Method 1 | | Method 2: Fusion | |
|---|---|---|---|---|---|---|---|---|---|---|---|
| | Xcept | Meso4 | RNet34 | EffB4 | f3net | SRM | SPSL | LBP | WDF | LFWS | LFWL |
| DFDCP | 0.9408 | 0.7937 | 0.9411 | 0.9313 | 0.9245 | 0.9380 | 0.9209 | 0.9213 | 0.9168 | 0.9262 | 0.9224 |
| DFDC | 0.6744 | 0.5729 | 0.6334 | 0.6890 | 0.6612 | 0.6856 | 0.6809 | 0.6713 | 0.6882 | 0.7061 | 0.6716 |
| FaceForensics++ | 0.6295 | 0.6132 | 0.6074 | 0.6462 | 0.6264 | 0.6404 | 0.6499 | 0.6274 | 0.6556 | 0.6464 | 0.6457 |
| FaceShifter | 0.4932 | 0.6246 | 0.4990 | 0.5419 | 0.5108 | 0.5237 | 0.5211 | 0.5519 | 0.5305 | 0.5561 | 0.5761 |
| DFD | 0.7401 | 0.6453 | 0.7004 | 0.7610 | 0.7062 | 0.7812 | 0.7515 | 0.7065 | 0.7294 | 0.7688 | 0.7385 |
| Celeb-DF-v1 | 0.6996 | 0.6504 | 0.6066 | 0.6583 | 0.6497 | 0.6486 | 0.6565 | 0.7134 | 0.7181 | 0.7488 | 0.7659 |
| Celeb-DF-v2 | 0.6873 | 0.6737 | 0.6928 | 0.6956 | 0.6812 | 0.7188 | 0.6797 | 0.7180 | 0.7297 | 0.7539 | 0.7651 |
| UADFV | 0.7757 | 0.8451 | 0.7710 | 0.8000 | 0.7669 | 0.8317 | 0.7937 | 0.8676 | 0.8749 | 0.8896 | 0.8903 |
| Avg AUC | 0.7051 | 0.6774 | 0.6815 | 0.7154 | 0.6909 | 0.7210 | 0.7068 | 0.7222 | 0.7304 | 0.7495 | 0.7470 |
| Improvement (%) | – | -2.77 | -2.36 | +1.03 | -1.42 | +1.59 | +0.17 | +1.71 | +2.53 | +4.44 | +4.19 |

- **Fusion methods generalize best.** Method 2 again produces the two strongest detectors: LFWS (0.7495), LFWL (0.7470).

- **Frequency-only gains shrink, fusion gains grow.** With changed dataset from FF++ to DFDCP, the relative improvements of F3Net/SRM change to –1.42 %/+1.59 %, whereas LFWS/LFWL rise to +4.44 %/+4.19 %. This indicates that pure frequency cues overfit the specific artifacts present in the training set, whereas the *learnable fusion block* still discovers complementary low- and high-frequency evidence that transfers.

- **Superior cross-domain robustness.** Although individual frequency-based baselines outperform the fusion models on certain datasets, the fusion block delivers *the highest **average** AUC across all tests*. This overall lead suggests that our method captures manipulation-invariant features that generalize across demographic variations and previously unseen forgery styles, rather than overfitting to dataset-specific artifacts.

When training datasets are matched in size but differ in style, the LFWS and LFWL models maintain or increase their performance advantage. This result demonstrates robustness that does not depend on the statistics of any single dataset.

**Why Not Train on Both FF++ and DFDCP?** In practice, jointly training on FF++ and the DFDCP datasets did not improve cross-domain performance. The observed degradation can be attributed to domain shifts, differences in collection conditions, and generation pipelines that introduce conflicting feature patterns. When these different datasets are combined, the model tends to focus on details specific to each dataset instead of learning features that work across domains. As a result, we test robustness within each dataset separately. This method is consistent with recent research showing that simply combining datasets can hurt performance in deepfake detection (Lai et al., 2024).

### 4.4 ABLATION STUDY: FROZEN FUSION BLOCK TRANSFERABILITY

We train LFWS and LFWL with Xception on FaceForensics++ (c23), freeze the fusion block, and attach it, without any retraining, to Meso-4 (MesoNet), ResNet-34, and EfficientNet-B4 (Tab. 3; Fig. 4).

**Backbone-wise analysis.** The frozen fusion block trained on Xception is reused directly, isolating its cross-backbone generalization. While both LFWS and LFWL consistently improve performance across backbones (see Tab. 3), we focus on the LFWS results for clarity. Xception achieves the highest improvement, with an increase of 3.8 in area under the curve (AUC). The frozen block processes feature statistics identical to those present during training. ResNet-34 demonstrates a 2.7 increase in AUC. The initial 3×3 convolutional layers and residual connections maintain the integrated feature representation, facilitating effective block transfer. EfficientNet-B4 shows a 0.9 increase in AUC. Because squeeze-and-excitation (SE) layers already perform channel re-weighting (Hu et al., 2018), further gains are small and only matter when accuracy changes of less than 1% are significant. Meso-4 (MesoNet) achieves a 1.0 increase in AUC. While aggressive down-sampling reduces the

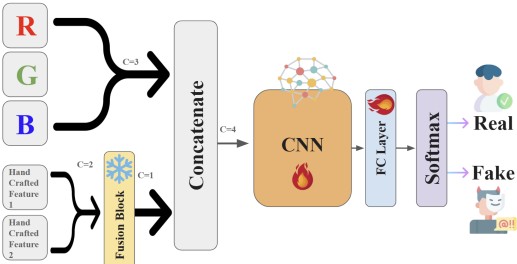

Figure 4: Frozen Fusion Block Inference Setup. The fusion block (trained with Xception on FF++) is frozen and reused as a preprocessing step for other backbones.

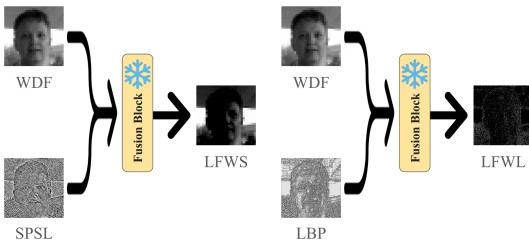

Figure 5: Visualization of fused features from our lightweight fusion block: in both LFWS and LFWL, the 1×1 conv assigns different weights—none are zero. LFWS blends WDF (+0.18) and SPSL (–0.12), while LFWL mixes WDF (+0.08) and LBP (–0.32). This shows that the fusion block learns how much to emphasize each feature, rather than discard them.

effect of the extra channel, adding 300 parameters still helps edge models with fewer than five million parameters.

The frozen fusion block is transferable across architectures, with the largest relative gains observed on mid-depth convolutional neural networks such as Xception and ResNet-34. These networks retain sufficient capacity to utilize an additional handcrafted channel without significantly increasing model size.

### 4.5 DISCUSSION AND OBSERVATIONS

**Lightweight Fusion vs. Direct Concatenation.** Appending handcrafted feature maps directly to RGB (Method 1) raises AUC over a vanilla CNN. Our lightweight block (Method 2) first compresses two handcrafted feature maps into one channel and then concatenates it with RGB, preserving a 4-channel input. Across eight benchmarks, Method 2 delivers the highest *average* AUC, even though it adds fewer parameters, showing that learned mixing of complementary cues is superior to naïve stacking. Specifically, LFWL combines WDF with LBP, while LFWS combines WDF with SPSL; each surpasses its Method 1 counterpart, confirming that the gain stems from better feature synergy rather than network size.

**Ablation Study1: Frozen Fusion Block Scalability.** We train the fusion block with Xception on FaceForensics++ (c23), freeze its weights, and deploy it unchanged on Meso-4 (MesoNet), ResNet-34, and EfficientNet-B4. All three backbones improve, proving that the block captures architecture-agnostic cues. The largest relative lifts appear in *mid-depth* networks, such as Xception and ResNet-34, which have sufficient capacity to exploit the extra channel without the diminishing returns observed in very deep EfficientNet-B4. Thus, a single, once-trained mixer generalizes broadly and scales gracefully.

**Ablation Study 2: Grad-CAM.** Figure 6 shows model attention on a forged frame. Xception attends broadly to the nose and mouth, while SPSL focuses on phase artifacts near the jaw. LBP highlights fine-grained textures around the lips. WDF, using low-frequency wavelet features, concentrates on the face boundary and hairline—regions where deepfake blending often introduces lighting and

Table 3: **AUC comparison across alternative backbones.** Baseline backbones without additional features are included for reference. The last two rows summarize the average AUC across all datasets and the relative improvement (%) over each backbone's baseline. Improvements are calculated as the percentage increase in average AUC compared to the baseline, demonstrating the added value of incorporating handcrafted features.

| Dataset | Baseline | | | | LFWL | | | | LFWS | | | |
|---|---|---|---|---|---|---|---|---|---|---|---|---|
| | Xcept | EffB4 | RNet34 | Meso4 | Xcept | EffB4 | RNet34 | Meso4 | Xcept | EffB4 | RNet34 | Meso4 |
| FaceForensics++ | 0.9805 | 0.9765 | 0.9665 | 0.7135 | 0.9741 | 0.9787 | 0.9605 | 0.6922 | 0.9761 | 0.9801 | 0.9616 | 0.6653 |
| FaceShifter | 0.6248 | 0.6062 | 0.5506 | 0.6275 | 0.6150 | 0.6002 | 0.5986 | 0.5904 | 0.6163 | 0.6381 | 0.6006 | 0.6031 |
| DFD | 0.8042 | 0.8535 | 0.7691 | 0.5524 | 0.8146 | 0.8387 | 0.7996 | 0.5671 | 0.8241 | 0.8385 | 0.7699 | 0.6235 |
| Celeb-DF-v1 | 0.6238 | 0.7363 | 0.6607 | 0.6022 | 0.7188 | 0.7262 | 0.7836 | 0.5383 | 0.7875 | 0.7238 | 0.7830 | 0.6310 |
| Celeb-DF-v2 | 0.6875 | 0.6684 | 0.6712 | 0.6080 | 0.7324 | 0.7430 | 0.7638 | 0.6135 | 0.7548 | 0.6958 | 0.7238 | 0.6175 |
| DFDCP | 0.6764 | 0.6737 | 0.6355 | 0.5734 | 0.7345 | 0.7078 | 0.6752 | 0.6744 | 0.6991 | 0.6753 | 0.6639 | 0.6255 |
| DFDC | 0.6862 | 0.6747 | 0.6932 | 0.5720 | 0.7072 | 0.6980 | 0.6610 | 0.5886 | 0.7172 | 0.6930 | 0.6527 | 0.5701 |
| UADFV | 0.8966 | 0.9072 | 0.9099 | 0.8714 | 0.9325 | 0.9454 | 0.8983 | 0.8691 | 0.9110 | 0.9205 | 0.9149 | 0.8637 |
| Avg AUC | 0.7475 | 0.7621 | 0.7321 | 0.6401 | 0.7786 | 0.7798 | 0.7676 | 0.6417 | 0.7858 | 0.7706 | 0.7588 | 0.6500 |
| Improvement (%) | – | – | – | – | +3.11 | +1.77 | +3.55 | +0.16 | +3.83 | +0.85 | +2.67 | +0.99 |

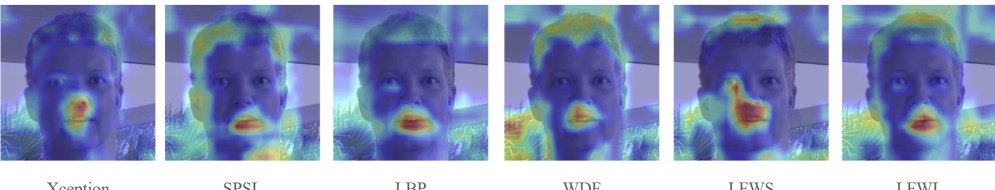

| Xception | SPSL | LBP | WDF | LFWS | LFWL |

Figure 6: Grad-CAM heatmaps on a forged frame. Xception focuses broadly on central facial regions, while SPSL and LBP highlight phase and texture inconsistencies near the jaw and lips. WDF emphasizes boundary and hairline areas where low-frequency blending artifacts often occur. Fusion-based models (LFWS, LFWL) yield more comprehensive attention across discriminative regions.

color inconsistencies. Fusing low-frequency and phase cues (LFWS) or combining with LBP (LFWL) produces more balanced and focused attention, demonstrating that frequency- and phase-domain features enhance anomaly localization.

## 5 CONCLUSION

The proposed *lightweight fusion block* combines two handcrafted streams: WDF and either LBP or SPSL cues. These are merged into a single-channel representation, which is then joined with the three RGB channels. This compact design helps LFWL and LFWS achieve the highest average AUC on all benchmark datasets. They perform better than deeper baselines and models that simply combine multiple channels. The results show that combining simple, learnable handcrafted features with RGB channels creates a strong and flexible base for handling different manipulations and capture conditions

## REPRODUCIBILITY STATEMENT

All the source code for our experiments is included as supplementary material. This code covers training, evaluation, and dataset. To maintain the anonymity of the review process and due to storage constraints, we have not included pre-trained model checkpoints at this stage.

## ETHICS STATEMENT

This research utilizes only publicly available face forgery datasets, including FaceForensics++, FaceShifter, DFD, Celeb-DF (versions 1 and 2), DFDCP, DFDC, and UADFV. We did not collect or

annotate any new human subject data, and we adhered to the licenses and usage conditions of each dataset. Our work is intended solely for improving deepfake detection and mitigating the potential harms of manipulated media, not for enabling or promoting the creation of such content.

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

## A APPENDIX

### A.1 LLM USAGE

Large language models (LLMs) were used only to provide limited help with minor language polishing and formatting. They were not involved in developing research ideas, designing experiments, analyzing data, or drawing conclusions. All scientific content and claims remain the work of the authors.

