# OpenReview forum: "Multi-Frequency Fusion for Robust Video Face Forgery Detection"
_ICLR.cc/2026/Conference — Submitted to ICLR 2026_

### Official Review · Reviewer_5zdC · 2025-10-29

**Soundness:** 1
**Presentation:** 2
**Contribution:** 1
**Rating:** 2
**Confidence:** 5

**Summary:**

The paper proposes a lightweight yet effective approach for face video forgery detection that challenges the prevailing trend of using large, wide, or dual-stream networks. Building upon the Xception baseline model, the authors introduce two variants LFWS and LFWL which integrate two handcrafted cues through a simple 1×1 convolutional fusion module. Experimental results demonstrate that the proposed method can effectively enhance the model’s generalizability on FF++ and DFDCP datasets.

**Strengths:**

This work proposes a lightweight plug-and-play feature fusion module which can be flexibly adapted to different backbones. The experimental results demonstrate the effectiveness and efficiency of the designed modules.

**Weaknesses:**

- The novelty of this work is limited. The use of SPSL, LBP, and WDF for deepfake detection has been extensively explored in prior studies, and the current method does not appear to introduce a distinct conceptual or methodological advancement.

- The experimental evaluation is restricted to only two datasets (DFDCP and FaceForensics++). Including additional datasets such as CDFv2, FFIW, and WDF would help better demonstrate the generalization and robustness of the proposed approach.

- The baseline comparisons are outdated. It is strongly recommended to include more recent state-of-the-art deepfake detection methods published in top-tier conferences and journals to provide a fair and comprehensive evaluation.

**Questions:**

NA

---

### Official Review · Reviewer_mcBT · 2025-10-30

**Soundness:** 1
**Presentation:** 1
**Contribution:** 1
**Rating:** 0
**Confidence:** 5

**Summary:**

Several critical flaws exist in this paper, detailed as follows:

* **Motivation issue:** In the abstract, the paper claims that existing methods mainly adopt *“wide or dual-stream”* architectures, motivating the proposal of a single, lightweight fusion model. However, this assertion is **inaccurate** and reflects a limited understanding of recent developments in this field. Many contemporary methods do not fall into this category — for example, **UCF (ICCV’23)**, **IID (CVPR’23)**, **LSDA (CVPR’24)**, **FreqBlender (NeurIPS’24)**, **ProDet (NeurIPS’24)**, **ForAda (CVPR’25)**, and **Effort (ICML’25)**.
  Furthermore, there is a **conceptual inconsistency** between the abstract and the introduction: while the abstract critiques “wide or dual-stream” designs, the introduction instead argues that prior methods focus on a single frequency band and thus motivates a multi-frequency fusion framework. These conflicting descriptions create confusion about the true motivation of the work. In addition, the paper provides **insufficient explanation or justification** for employing *Local Binary Patterns (LBP)* and *Spatial-Phase Shallow Learning (SPSL)* as core components of the method.

* **Novelty issue:** The proposed framework appears to be a **straightforward combination** of two well-established strategies — LBP and SPSL — without offering new insights or deeper exploration into their underlying principles. As a result, the novelty of the method is quite limited.

* **Experimental issue:** The experimental evaluation is **outdated and incomplete**. The comparisons are made only against a few basic models such as **Xception, Meso4, RNet34, EffB4, F3Net, SRM,** and **SPSL**. These baselines are outdated, and also insufficiently cited. Based on common knowledge, most of these methods are several years old, and thus the results fail to convincingly demonstrate the effectiveness of the proposed approach against **modern baselines**.

* **Writing and organization issue:** The **related work section** is superficial and fails to cover key representative studies in this area. Moreover, including a motivation paragraph within the related work section is **inappropriate** and disrupts the logical flow. The overall **organization of text, figures, and explanations** is also problematic, with several expressions lacking clarity and precision.

Overall, the paper suffers from serious issues in motivation consistency, originality, experimental validation, and writing quality, which substantially weaken its credibility and contribution.

**Strengths:**

NA

**Weaknesses:**

See summay parts

**Questions:**

NA

---

### Official Review · Reviewer_xDMZ · 2025-10-31

**Soundness:** 3
**Presentation:** 2
**Contribution:** 2
**Rating:** 4
**Confidence:** 3

**Summary:**

This paper proposes a highly efficient approach for deepfake detection by fusing handcrafted features from different frequency domains Based on the Xception model. The core idea is that while existing detectors often focus on either high-frequency artifacts (e.g., SPSL, LBP) or low-frequency structural cues (e.g., Wavelet-Denoised Features, WDF), combining them in a lightweight, learnable manner yields superior robustness.The authors introduce two specific models, i.e., LFWS and LFWL to deal with it.

**Strengths:**

1.The paper's key strength is a notable 3.8-4.4% AUC gain over the Xception baseline with only 292 additional parameters.

2.The experimental setup is thorough and credible

3.This paper proposes well-designed ablation studies

**Weaknesses:**

1.The paper only focuses on fusing WDF with either SPSL or LBP. It does not explore fusing SPSL and LBP together (both high-frequency features) or a three-way fusion of WDF, SPSL, and LBP. While the chosen pairs are well-motivated (low + high frequency), the absence of these other combinations leaves the question of whether the presented pairs are optimal

2.The benchmarks mainly include face-swaps and reenactments, leaving the method's robustness to tougher real-world threats like compressed videos, adversarial attacks, or advanced diffusion-model forgeries untested.

3.The lack of cross-domain improvement from joint training on FF++ and DFDCP raises concerns about the method's claimed robustness, especially since the authors offer no evidence or analysis for their "conflicting feature patterns" explanation.

**Questions:**

See weaknesses

---

### Official Review · Reviewer_fx9s · 2025-11-01

**Soundness:** 2
**Presentation:** 3
**Contribution:** 2
**Rating:** 2
**Confidence:** 5

**Summary:**

This paper proposes a lightweight, multi-frequency fusion framework for video face forgery detection. The authors argue that existing detectors are either too computationally heavy or focus on a single frequency band. The proposed method builds on an Xception baseline by introducing a minimal fusion block (a 1x1 convolution adding only 292 parameters) to combine a low-frequency feature (Wavelet-Denoised Feature, WDF) with a high-frequency feature (either Spatial-Phase Shallow Learning, SPSL, or Local Binary Patterns, LBP). The resulting models, LFWS (WDF+SPSL) and LFWL (WDF+LBP), are evaluated on eight public benchmarks. The authors claim these models achieve state-of-the-art robustness, demonstrating significant AUC gains over the Xception baseline and outperforming several previous frequency-based methods like F3Net and SRM.

**Strengths:**

1. The paper is well-written, clearly structured, and easy to follow.
2. The primary methodological strength is the minimal computational overhead; the fusion block adds only 292 parameters.
3. The ablation study in Section 4.4 (Table 3) is valuable. It demonstrates that the fusion block, trained once on Xception, can be frozen and transferred to other backbones (like ResNet-34 and EfficientNet-B4) as a plug-and-play module, providing a consistent performance boost .

**Weaknesses:**

1. The core technical contribution, the "Lightweight Fusion Block," is a standard 1x1 convolution followed by Batch Normalization and ReLU. This is a ubiquitous component, not a novel methodology. The paper's reliance on a "handcrafted features + simple CNN" paradigm is outdated for a top-tier venue, as it lacks methodological depth.
2. The method's reliance on WDF, LBP, and SPSL makes it inherently brittle. These features are designed to detect known artifacts (e.g., texture anomalies, phase distortions). This approach is a static target; future generative models will inevitably learn to circumvent the detection of these specific, pre-defined flaws, rendering the method obsolete.
3. The paper's SOTA claims are invalid. It ignores 4-5 years of progress in the field (e.g., Transformer-based detectors, universal forensic methods) and only compares against obsolete methods from 2020-2021 (F3Net, SRM). This makes the performance value of the contribution impossible to ascertain.
4. The introduction correctly identifies "diffusion models" as a key challenge. However, the experimental evaluation contains zero diffusion-based benchmarks, relying exclusively on legacy GAN and classic forgeries. The method is not validated against the modern threats it uses as motivation.

**Questions:**

1.Can the authors justify the complete omission of any SOTA baselines from the 2022-2025 period? Please provide a comparison against modern Transformer-based (e.g., ViT) or universal forensic detectors.

2.Given that diffusion models are a primary motivation, can the authors provide experimental results on any modern diffusion-based deepfake benchmark? How can the robustness claims be evaluated without this?

3.Please elaborate on the performance degradation observed during joint training (FF++ and DFDCP). How is this finding reconciled with the paper's central claim of improved robustness and generalization?

4.What is the long-term viability of a method based on static, handcrafted features in an adversarial field where generators are constantly evolving to eliminate the very artifacts this method targets?

5.The "Method 1" (concatenation) baseline in Table 1 only tests RGB + one feature (4 channels total). A fairer comparison to "Method 2" (RGB + 1 fused channel) would be to concatenate both features (e.g., RGB + WDF + SPSL, for a 5-channel input). Was this more direct, multi-channel concatenation baseline evaluated?

---

### Meta-Review · Area_Chair_nmou · 2026-01-03

**Summary:**

The reviewers agree that the paper proposes a computationally efficient framework for video face forgery detection by fusing handcrafted frequency-domain features using a lightweight fusion block. While the efficiency and plug-and-play nature of the design are acknowledged, the reviewers raise substantial concerns regarding the limited novelty of the method, outdated experimental comparisons, and insufficient validation against modern forgery threats. Several reviewers also point out inconsistencies and weaknesses in the paper’s motivation and positioning, as well as shortcomings in experimental design and coverage. Overall, the reviewers express significant doubts about whether the current contribution meets the methodological and empirical standards expected at ICLR.

**Reviewer Concerns:**

No rebuttal or author response was provided, and there was no reviewer discussion. As a result, all concerns raised in the original reviews regarding novelty, experimental evaluation, motivation consistency, and robustness remain unaddressed.

**Reviewer Scores:**

Since no rebuttal or discussion took place, reviewers would be expected to maintain their original scores, as there was no additional information or clarification to prompt any reassessment.

---

### Decision · Program_Chairs · 2026-01-26

Reject